# Knowledge attitudes and practices toward seasonal influenza vaccine among pregnant women during the 2018/2019 influenza season in Tunisia

**Sonia Dhaouadi**[1], **Ghassen Kharroubi**[2,3], **Amal Cherif**[1], **Ines Cherif**[2,3], **Hind Bouguerra**[1], **Leila Bouabid**[1], **Nourhene Najar**[1], **Adel Gharbi**[2,3], **Afif Ben Salah**[2,3,4], **Nissaf Bouafif ép Ben Alaya**[1], **Jihene Bettaieb**[2,3]*

1 National Observatory of New and Emerging Diseases, Tunis, Tunisia, 2 Laboratory of Medical Epidemiology, Pasteur Institute of Tunis, Tunis, Tunisia, 3 Laboratory of Transmission, Control and Immunobiology of Infections (LR11IPT02), Pasteur Institute of Tunis, Tunis, Tunisia, 4 Arabian Gulf University, Manama, Bahrain

* bettaiebjihene@yahoo.fr, jihene.bettaieb@pasteur.rns.tn

**Data Availability Statement:** All relevant data are within the manuscript and its Supporting Information files.

## Abstract

### Background

The uptake and acceptance of the influenza vaccine (IV) among pregnant women remain unknown in Tunisia despite the increased influenza-related complications and death. The present study aimed to assess the IV uptake and acceptability and to describe related knowledge and attitudes among pregnant women in Tunisia.

### Methods

A cross-sectional study was conducted in 84 Tunisian healthcare facilities over a period of three months (from March to May 2019). All pregnant women aged ≥18 years who sought antenatal care in related health structures were included in this study based on a multistage self-weighted sampling. We measured knowledge and attitudes towards the IV and assessed factors related to willingness for its uptake.

### Results

The questionnaire was completed by 1157 pregnant women. More than half of the participants (60.2%; 95% confidence interval [CI] [57.3%–63.0%]) reported awareness about the IV. Among included PW, only 4.6%; 95% CI [3.5%–6.1%] received it during their current pregnancy. However, (36.8%; 95% CI [34.0%–39.6%]) declared their willingness to receive the vaccine in the next pregnancy. Recommendation by healthcare providers, identified to be the most trustful source of information, was the main reason for acceptance. However, the intention to accept the IV by pregnant women was significantly associated with such recommendation and perceived safety and effectiveness of this vaccine.

**Funding:** This project was funded by the Task Force for Global Health [grant numbers: F821E23E-52C0-4C51-8FC9-C356C34499B2].

**Competing interests:** We have no conflicts of interest to disclose.

## Conclusion

Antenatal care visits are a precious opportunity that should not be missed by health care providers and especially gynecologists to promote the IV uptake by pregnant women in Tunisia.

## I. Introduction

Influenza is an acute respiratory infection caused by viruses belonging to the Orthomyxoviridae family. The burden of this disease varies based on characteristics of the circulating strains and the immune status of the population [1].

Pregnant women (PW) have been identified as a group at an increased risk of influenza-related complications and death, during seasonal epidemics and pandemics, due to physiological and immunological changes [2–5]. Furthermore, influenza during pregnancy has been reported to have considerable effects on the fetus and is associated with an increased risk of early spontaneous abortion, neurological birth defects, preterm delivery, stillbirth, low birth weight, and neonatal death in severe cases, especially among babies born during influenza season [2,3].

Several reports have supported the safety, immunogenicity, and effectiveness of the inactivated influenza vaccine (IV) (trivalent inactivated IV) during any term of pregnancy for the mother and the fetus [6–8]. In fact, maternal antibodies produced after vaccinations are transferred to the fetus and newborn via the umbilical cord, placenta, and breastfeeding [9]. The cost effectiveness of the IV was confirmed by several studies [10,11]. Considering these reasons, since 2012, the World Health Organization (WHO) has declared that PW should be given the highest priority if countries consider initiation or expansion of their seasonal influenza vaccination programs [3].

Despite the proven effectiveness of the IV in protecting the mother, fetus, and new born, the rate of maternal influenza immunization remains low in many countries [12–15], and significantly lower than the target of 80% fixed by authorities in 2020 for healthy individuals [16].

In Tunisia, the IV is not free of charge among pregnant women and non-refundable. The uptake and the acceptance of IV among PW remain unknown. The present study was thus conducted to assess IV uptake and acceptability among PW and to describe their knowledge and attitude regarding this vaccine. The findings from this research will constitute an evidence base for the promotion of the IV by the health authorities in charge of the national program of seasonal influenza control and prevention in Tunisia.

## II. Methods

### 1. Study design

A nationwide cross-sectional study was conducted in Tunisia during the 2018/2019 influenza season over a period of three months (from March to May 2019, spring season). The 2018/2019 influenza season in Tunisia began in week 40/2018 (1$^{st}$ October, 2018) and ended in week 18/2019 (May,5$^{th}$, 2019).

### 2. Study population

All PW aged ≥18 years who attended a public health center (primary healthcare center, district or regional hospital) for routine antenatal care during any term of pregnancy were included in this survey. Women suffering from cognitive disorders were excluded.

The total population of PW who attended the primary and secondary health care facilities is estimated to be between 10,000 and 100,000.

The calculated sample size was ≃1200 PW after adjustment based on a non-response rate of 20% and a design effect of 2.5.

## 3. Sampling procedure

A multistage self-weighted sampling method was used to select PW. First, a stratified sampling was performed based on the three Tunisian regions (North, Center and South). Given the limited budget of the survey (limited logistics support and resources), we chose to randomly select one third (n = 8) of the total 24 Tunisian governorates.

We applied the percentage of Women of childbearing age in each region (North, Center and South) to determine the number of selected governorates in each region. Thus, we selected the following governorates in each region (S1 Fig):

- Four in northern Tunisia (48.0%*8): Ariana, Ben Arous, Bizerte, Siliana

- Three in the central Tunisia (36.7%*8): Sousse, Mahdia, Kairouan

- One in southern Tunisia (15.3%*8): Gafsa

The second stage consisted on a stratification by area of residence (urban and rural).

The third stage was the selection by simple random sampling in each governorate of health care centers which offer antenatal care visits from both urban and rural areas. The selection of centers was performed as the survey progressed. Every day, we randomly selected one center among all those that offer medical care consultations for PW on that day. The selected center will not be selected again.

On the day of their visit, investigator(s) were asked to randomly select participants from the exhaustive list of PW consulting the selected health care facility on that day. PW were ordered by order of registration, then they were randomly selected using a random number generator. The number of PW approached to participate to the survey in each health care facility depended on the availability of human and logistic resources.

It should be noted that a pilot study was performed in order to assess the comprehensiveness of questionnaire items, to identify ambiguous questions and to estimate the maximum number of questionnaires that could be properly administrate by one investigator which was equal to 20.

The selection of PW was stopped when we achieved the required sample size.

The calculated sample size was distributed according to the distribution of women of childbearing age between the three Tunisian regions, to the weight of each governorate in the corresponding region and to the weight of the area of residence in each governorate.

More details are presented in S1 Fig.

The survey was conducted in 84 health care facilities: Primary healthcare centers (n = 69; 82%), district hospital (n = 11; 13%) and regional hospital (n = 4; 5%) (S1 and S2 Tables)

## 4. Data collection

A face-to-face interview was conducted using a standardized questionnaire by interviewers previously trained for this purpose. The interviewers were health care professionals (medical doctor, nurse and midwife) working in public health institutions.

The questionnaire (S1 and S2 Appendix) included 38 items and was divided into four parts: socio-demographic characteristics, pregnancy-related data, knowledge, and attitude regarding influenza and uptake of the IV. A woman was considered to have been vaccinated against influenza at the time of the survey if she reported having received one dose of influenza vaccine during this pregnancy (independent of the term of pregnancy).

Responses to general statements about knowledge and attitude regarding influenza (one statement) and IV (seven statements) were measured using a 5-point Likert scale ranging from strongly disagree (1) to strongly agree (5).

## 5. Data analysis

EpiInfo software version 7.2.2.6 (developed by Centers for Disease Control and Prevention, U. S) was used for data entry and analyses.

This study included only qualitative variables that were presented as numbers and percentages.

Answers to questions assessed using the 5-point Likert scale were recorded as follows: "strongly agree" responses were combined with the "agree" responses and "strongly disagree." "Disagree," "neither agree nor disagree," and "I don't know" responses were combined into "other" responses.

Percentages and their 95% confidence interval were weighted according to region, governorate and area of residence.

Pearson's χ2 test was used for bivariate analysis to assess any associations between willingness to receive the IV among PW during their next pregnancy (variable of interest) and sociodemographic and pregnancy characteristics. History of IV uptake, vaccine uptake during the current pregnancy, knowledge and attitudes about influenza infection and IV were also tested. We calculated weighted crude odds ratios (OR) and their 95% CI according to region, governorate and area of residence in order to assess strength of association between variables.

Differences were considered as statistically significant if the p-value was less than 0.05.

## 6. Ethical considerations

Ethical approval for this survey was obtained from the Biomedical Ethics Committee of the Pasteur Institute of Tunis. Approval was also obtained from the Tunisian Ministry of Health.

All women included in the survey were informed about the objectives and modalities of the study. Women who agreed to be interviewed provided written informed consent. Participants were also informed that all data collected would be analyzed anonymously and that participation was voluntary.

A unique ID was assigned to each participant, and the hard copy of the questionnaires was stored in a secure location, with access restricted to approved survey personnel.

## III. Results

### 1. Characteristics of the included pregnant women

Overall, 1348 PW attending 84 health care centers were approached to participate in the survey. Among them, 1200 PW accepted to respond to our questionnaire. However, 1157 questionnaires were retained for analysis and 43 were excluded for incomplete data or the presence of an exclusion criterion (age less than 18 years). The response rate was 89% (1200/1348).

More than half (74.7%) of the surveyed PW resided in urban areas. Most women (62.4%) were between 25 and 34 years of age and less than a third had attended university (23.2%).

Regarding pregnancy history, 48.9% and 41.3% were in the second and third trimester of pregnancy, respectively. Nearly two women of five (41.5%) were multigravida and 31.4% reported that the current pregnancy was their first. Furthermore, 17.8% of the surveyed women had comorbidities and 21.0% reported complications prior to or in the current pregnancy, respectively (Table 1).

**Table 1. Socio-demographic and pregnancy characteristics of the enrolled women, (n = 1157).**

| Variables | n | Percentage (%) |
|---|---|---|
| Sociodemographic characteristics | | |
| Residence area (n = 1157) | | |
| Urban | 864 | 74.7 |
| Rural | 293 | 25.3 |
| Age groups (years) (n = 1157) | | |
| 18–24 | 146 | 12.6 |
| 25–34 | 722 | 62.4 |
| ≥35 | 289 | 25.0 |
| Educational level (n = 1152) | | |
| Primary school or less | 293 | 25.4 |
| Secondary school | 590 | 51.2 |
| University level | 267 | 23.2 |
| Don't want to answer | 2 | 0.2 |
| Marital status (n = 1136) | | |
| Single | 4 | 0.4 |
| Married | 1126 | 99.0 |
| Others situation | 3 | 0.3 |
| Don't want to answer | 3 | 0.3 |
| Employment status (n = 1140) | | |
| Employed | 387 | 33.9 |
| Unemployed | 752 | 66.0 |
| Don't want to answer | 1 | 0.1 |
| Time required to go from home to antenatal care facility (minutes) (n = 1146) | | |
| ≤30 | 929 | 81.1 |
| 31–60 | 185 | 16.1 |
| ≥61 | 29 | 2.5 |
| Don't know | 3 | 0.3 |
| Pregnancy related characteristics | | |
| Number of pregnancies (n = 1148) | | |
| 1 (this is my first pregnancy) | 361 | 31.4 |
| 2 | 311 | 27.1 |
| ≥3 (multigravida) | 476 | 41.5 |
| Trimester of this pregnancy (n = 1146) | | |
| First Trimester (1–13 weeks) | 112 | 9.8 |
| Second Trimester (14–26 weeks) | 560 | 48.9 |
| Third Trimester (≥27 weeks) | 474 | 41.3 |
| Number of antenatal care visits completed by time of survey | | |
| 1 (this is my first antenatal consultation) | 109 | 9.5 |
| >1 | 1040 | 90.5 |
| Comorbidities prior to this pregnancy (n = 1149) | | |
| Yes | 205 | 17.8 |
| No | 944 | 82.2 |
| Complications during this pregnancy (n = 1137) | | |
| Yes | 239 | 21.0 |
| No | 898 | 79.0 |
| Number of children (n = 1087) | | |
| 0 | 372 | 34.2 |

(*Continued*)

**Table 1.** (Continued)

| Variables | n | Percentage (%) |
|---|---|---|
| 1–2 | 604 | 55.6 |
| ≥3 | 111 | 10.2 |
| History of miscarriage, abortion, stillbirth (n = 1148) | | |
| Yes | 297 | 25.9 |
| No | 851 | 74.1 |

## 2. Influenza vaccine uptake

Among the surveyed PW, 988 had heard of influenza in the past (86.0%; 95% CI [83.9%–87.8%]) and 694 already know about the IV (60.2%; 95% CI [57.3%–63.0%]). Seventy-five (6.7%; 95% CI [5.4%- 8.4%] of PW reported having enough information about side effects of IV.

Among included PW, 78 (7.0%; 95% CI [5.6%–8.6%]) women had received the IV at least once in the past and only 51 (4.6%, 95% CI [3.5%–6.1%]) received the vaccine during the current pregnancy. Seventy-five women (6.7%; 95% CI [5.4%–8.4%]) reported being vaccinated against influenza at least once in the past 5 years.

In addition, 550 (48.4%; 95% CI [45.6%–51.3%] of interviewed women reported that they would accept the vaccination if recommended by the health professionals and offered free of charge.

## 3. Factors associated with influenza vaccine acceptance in the next pregnancy

Among the included women, 421 (36.8%; 95% CI [34.0%–39.6%]) were willing to receive the IV in their next pregnancy and 154 (14.1%; 95% CI [12.1%–16.4%]) answered "I don't know."

The results of bivariate analysis revealed that both of women with comorbidities prior to the current pregnancy and those with complications during the current one were more likely to be willing to receive the IV (OR = 1.9; 95%CI [1.3–2.6], $p<10^{-3}$ and OR = 1.5; 95%CI [1.1–2.1], p = 0.01 respectively). However, neither the number of antenatal care visits completed at the time of the survey, nor the number of pregnancies, and number of children were significantly associated with the willingness to receive the vaccine (S3 Table).

As presented in Table 2, PW who received the IV at least once in the past were 2.7fold more likely to retake it compared to those who were never vaccinated (OR = 2.7; 95% CI [1.6–4.5]; $p<10^{-3}$).

In addition, participants who were unaware of the IV or adverse events reported by persons who received the IV were 2 times (OR = 1.9; 95% [1.5–2.6]; $p<10^{-3}$) and 2.4 times (OR = 2.4; 95% CI [1.2–4.8]; p = 0.01) more likely to be willing to accept the vaccine respectively.

Similarly, women who had adequate information regarding the safety and side effects, those who trusted the advice of their healthcare provider, and those who were recommended to receive IV during their pregnancy were more likely to accept the IV (Table 2).

Compared to PW who refused the IV, women who were willing to receive it were more convinced that the vaccine helped in protecting them as well as their fetus, and newborn ($p<10^{-3}$). They also perceived the vaccine harms as significantly reduced. More details are presented in Table 3.

Healthcare workers were identified as the most trusted source for information on the IV: doctors (87.7%) and other healthcare professionals (nurse, midwife, and pharmacist, among others) in 93.3% of the cases. Furthermore, 74.5% and 34.0% of the surveyed women reported

**Table 2. Knowledge and attitudes towards influenza and influenza vaccination and their association with willingness to receive influenza vaccine during pregnancy [1].**

| Variables[2] | Willing to receive influenza vaccine n (%)* | Crude OR 95% CI[3]* | p-value |
|---|---|---|---|
| Do you know anyone who has been severely ill with influenza? | | | 0.7 |
| No (n = 632) | 272 (43.9) | 1 | |
| Yes (n = 325) | 136 (42.6) | 1.0 [0.7–1.3] | |
| Had you heard of the influenza vaccine before?(Q15) | | | <10⁻³ |
| Yes (n = 619) | 222 (36.6) | 1 | |
| No (n = 373) | 196 (52.9) | 1.9 [1.5–2.6] | |
| Do you know anyone who had a bad reaction to influenza vaccine? | | | 0.01 |
| Yes (n = 51) | 12 (24.8) | 1 | |
| No (n = 919) | 398 (43.8) | 2.4 [1.2–4.8] | |
| Had you enough information about safety and side effects of influenza vaccine? | | | 10⁻³ |
| No (n = 870) | 353 (41.0) | 1 | |
| Yes (n = 73) | 45 (61.5) | 2.3 [1.4–3.8] | |
| Do you trust the advice of your health care provider? | | | 0.002 |
| No (n = 81) | 23 (27.3) | 1 | |
| Yes (n = 853) | 387 (46.2) | 2.3 [1.3–3.9] | |
| Has anyone recommended you receive influenza vaccine during this pregnancy? | | | <10⁻³ |
| No (n = 890) | 359 (40.8) | 1 | |
| Yes (n = 92) | 56 (61.2) | 2.3 [1.5–3.6] | |
| Has anyone discouraged you from receiving influenza vaccine during this pregnancy? | | | 0.3 |
| Yes (n = 81) | 30 (36.8) | 1 | |
| No (n = 895) | 382 (43.3) | 1.3 [0.8–2.1] | |
| Did you receive the influenza vaccine at least once at the past? | | | <10⁻³ |
| No (n = 919) | 371 (40.8) | 1 | |
| Yes (n = 72) | 46 (65.1) | 2.7 [1.6–4.5] | |
| Influenza vaccine uptake during this pregnancy | | | 0.09 |
| No (n = 946) | 393 (42.0) | 1 | |
| Yes (n = 45) | 24 (55.3) | [0.9–3.2] | |

*: Weighted according to region, governorate and area of residence.

[1]Persons who answered 'I do not Know' to the question concerning willingness to receive influenza vaccine were excluded (n = 154)

[2]respondent indicated "I don't know" or "I don't remember" or skipped the question are not included in Table 2 calculations.

[3] Confidence interval.

that they would accept the vaccination if recommended by the medical doctor and other health provider respectively. The main three reasons for refusing IV were fear of side effects and concerns regarding self-harm or harm to the fetus (72.1%), concerns about vaccine efficacy (16.2%) and natural immunity inducted by the infection is better than the immunity inducted by the vaccine (11.2%).

**Table 3. Comparative analysis of knowledge and attitudes about influenza infection and influenza vaccination according to willingness to receive influenza vaccine during pregnancy.**

| Statements[1] | Willing to receive influenza vaccine | | Not willing to receive influenza vaccine | | |
|---|---|---|---|---|---|
| | N[2] | Agree[3] n (%) | N[4] | Agree[3] n (%) | p-value |
| • Influenza is more dangerous for pregnant women than no pregnant women (n = 835) | 344 | 295 (**85.8**) | 491 | 417 (84.9) | 0.74 |
| • Influenza vaccine can be dangerous for pregnant women (n = 611) | 218 | 67 (30.7) | 393 | 169 (**43.0**) | 0.003 |
| • Influenza vaccine can be dangerous for the fetus (n = 612) | 220 | 58 (26.4) | 392 | 159 (**40.6**) | $10^{-3}$ |
| • Influenza vaccine can be dangerous for the newborn (n = 609) | 217 | 54 (24.9) | 392 | 149 (**38.0**) | $10^{-3}$ |
| • Influenza vaccine helps protect pregnant woman against influenza (n = 603) | 219 | 162 (**74.0**) | 384 | 128 (33.3) | $<10^{-3}$ |
| • Vaccination of pregnant women against influenza helps protect the fetus (n = 597) | 214 | 133 (**62.1**) | 383 | 86 (22.5) | $<10^{-3}$ |
| • Vaccination of pregnant women against influenza helps protect the newborn (n = 602) | 217 | 124 (**57.1**) | 385 | 82 (21.3) | $<10^{-3}$ |
| • Women should receive the Influenza vaccine during each pregnancy (n = 602) | 215 | 133 (**61.9**) | 387 | 45 (11.6) | $<10^{-3}$ |

[1]Statements about influenza concern only participants who had knowledge about influenza while statements about influenza vaccine concern only participants who know that there exists a vaccine.

[2] The number of responses for each statement among participants that indicated a willingness to receive influenza vaccine.

[3]Agree also includes strongly agree".

[4] The number of responses for each statement among participants that indicated unwillingness to receive influenza vaccine.

## IV. Discussion

Similar to the findings of our study, other studies have reported very low vaccine uptake during pregnancy, mainly in developing countries. The low vaccine uptake observed in the present study (4.6%; 95% CI [3.5%-6.1%]) may be explained by the absence of a national policy regarding flu immunization among PW. In fact, in Tunisia, IV is provided free of charge only to the elderly with underlying chronic diseases and healthcare workers in public health settings. Furthermore, the national health insurance does not cover the cost of the vaccine. This low coverage rate seems be related to the lack of awareness of PW regarding the need of IV during pregnancy and of physician recommendation of IV rather than IV availability and health care accessibility. In addition, Tunisia (upper-middle income country) appeared among the only six countries in Africa that adopted national immunization policies or guidelines against seasonal influenza (National program for surveillance and control of influenza since 1980 and Pandemic influenza preparedness and response plan since 2009) and more than 90% of Tunisian population resided less than 5 Km of a primary health care center [17,18]. For instance, in a study conducted in Thailand during the 2012–2013 influenza season, only 4% of the surveyed PW had received the vaccine during their pregnancy [12]. Likewise, according to an administrative estimation of vaccine coverage, only 1% of pregnant Moroccan women were vaccinated in 2016 [19]. Li Richun et al. reported that none of the PW interviewed in a Chinese qualitative study had received the vaccine during the 2015–2016 influenza season [14].

This could be explained by the difference between public health national program goal between countries. In high income countries (HIC), policies are based on a variety of measures, including cost-effectiveness, prevention of work and school absenteeism, prevention of ambulatory care visits, hospitalizations, and death. In low and middle income countries (LMIC), strategies are mainly focused on vaccine-attributable severe disease prevention and program costs. Among the several challenges to influenza vaccine program implementation in those countries (National immunization programs), most notably were limited disease burden data, lack of awareness about influenza disease burden among stakeholders, uncertain impact (benefits and safety) of IV on important public health outcomes, and technical challenges providing vaccination services [20,21]. In fact, higher coverage rate was not correlated with the

level of economic development while the influenza management was enhanced when the vaccine was offered free through the public sector [22].

According to a systematic review conducted by WHO, increasing seasonal influenza coverage in the LMIC must overcome several challenges: First, lack of information on influenza disease burden, vaccine effectiveness and impact, a poor definition of individuals at risk of influenza-related complications, and a low perceived severity of influenza disease, places seasonal influenza vaccination low on the list of a country's public health priorities.

Second, a lack of a cogent immunization policy, high vaccine costs coupled with limited resources for supply and delivery along with public adverse opinions against vaccination limit wider usage of the vaccine in those countries. Third, the low vaccine demand and poor coverage further contributes to the already poor or absent vaccine production capacity to make sustainability a major challenge for the influenza vaccine industry in LMIC [17].

However, the coverage was higher in some developed countries such as the United States (61.2% in November 2019), the United Kingdom which vaccine coverage ranged from 44.1% in England to 56.1% in Northern Ireland in 2014–15 and was 45.2% in 2018–2019 influenza season (from 44.3% in Northern Ireland to 74.2% in Wales) [23–27]. This high flu vaccine uptake could be explained by that the predominant payment mechanism for influenza IV was through the national health services in the European member states and by the implementation of a vaccine coverage monitoring system in this target group. In the UK, the target PW are listed and invited to get the vaccine. In addition, obstetrician-gynecologists doctor closely worked with community midwives to ensure accurate and timely recording updates of PW vaccinated outside general practice [28]. In the USA, coverage rates seem to be higher when a healthcare provider can recommend, offer, and administer the vaccine at the same visit as opposed to making a recommendation and referring the patient elsewhere to receive the vaccine [28].

Those results could be thus explained by the seniority of implementation of national vaccination program among high risk groups (of which PW) which allow to monitor, to evaluate and to remedy deficiencies in high income counties. Unlikely to barriers to the effective implementation of the national immunization strategy among general population and PW in particular in LMIC.

Among the total participants, 36.8% reported their willingness to receive the IV. This proportion was significantly higher than the vaccine uptake assessed among the surveyed PW. However, the proportion was lower than the acceptance rates of the IV reported in other studies: 42% in Thailand, 64.5% in Taiwan, 87% in Pakistan, and 76.3% in China [12,29–31]. Nevertheless, it is important to note that differences in the study populations and methods used while assessing acceptance of the IV may hinder comparison between studies.

The results of our study highlighted that only 48% of women claimed that they would take the vaccine if it was recommended by a health care professional and provided free of charge, which is substantially lower than the target of 80% vaccinated among PW, even if this would be a large improvement from the 4% of PW who reported IV uptake during the current pregnancy. Moreover, three out of four women (75%) declared that they would be willing to take it if it was recommended by a medical doctor, but only 34% if it was recommended by another health care professional.

Prior comorbidities and complications during the current pregnancy were positively associated with willingness to receive the vaccination. This could be explained by a higher perception of vulnerability among this population compared to PW without a medical history.

In addition, we found that prior recommendation to receive IV was significantly associated with a higher rate of acceptance. Similar results were reported by Ditsungnoen et al andOffedu et al [12,32,33]. Furthermore, IV recommendation by healthcare workers was reported as the

most important reason for acceptance, and caregivers were identified as the most trusted source of information, as reported in other studies as well [34–36]. Such findings emphasize the role of healthcare professionals in improving the acceptance rate of IV among PW. Therefore, efforts should be paid to increase awareness among healthcare providers regarding the importance of maternal immunization against influenza, and to encourage PW to be vaccinated.

Consistent with the observations of other studies [12,14,29,30], PW who believed in the effectiveness of the IV for the mother, fetus, and newborn were more inclined to be vaccinated. However, concerns regarding dangers of the vaccine to the mother, fetus, and newborn child were negatively associated with willingness to such vaccine as reported in previous studies [37,38]. Clarifications should thus be made concerning vaccine benefits and innocuousness during pregnancy through regular health education programs. In fact, only 6.5% of the participants declared having adequate information on the safety and side effects of the IV.

We also observed, in concordance with other studies [12,29], that a history of IV was significantly associated with willingness to retake the vaccine. Among the PW who had been vaccinated at least once in the past, 65.1%intended to uptake the vaccine during their next pregnancy. These findings suggested that a successful experience of influenza immunization encouraged PW to continue receiving the vaccine. We observed that awareness about IV-related adverse events was negatively associated with willingness to receive the vaccine. Our results showed that PW who believe that the IV could protect both mother and fetus were more likely to accept to receive the vaccine in the next pregnancy. This may be explained by the spread of rather false information regarding the vaccine. Thus, health care professionals should take the lead in the awareness regarding benefits of the influenza vaccination among pregnant woman in the context of properly designed and implemented health education programs. Poor knowledge and negative attitude towards seasonal IV influence the acceptance of IV. Therefore, health care providers' recommendations are important to PW's acceptance of IV. Health education, direct and regular communication strategies on IV and influenza infection are necessary to enhance the acceptance of IV vaccination among this high risk group. Drivers of the latter behavior seem to fit the classical framework of the health belief model [31,39].

## Strengths and limitations of study

To the best of our knowledge, this is the first nationwide study of its kind in Tunisia. Moreover, unlike other descriptive studies that have a primarily exploratory purpose, the knowledge, attitudes, and practices (KAP) surveys focus on the problem that is being addressed as well as the means that could facilitate its understanding and resolution. KAP surveys chiefly have an interventional purpose, aiming at optimizing practices and promoting health [40]. Therefore, this survey enabled us to derive, for the first time in Tunisia, benchmark information on the knowledge and perceptions regarding the vaccination among this high priority group, as well as data regarding uptake of influenza immunization in PW. The results provided the fundamental information required to develop appropriate recommendations to optimize acceptance and coverage of the vaccination among PW in our country. Our survey was based on a representative sample of all PW aged 18 and over and attending primary and secondary health care centers from public health sector in Tunisia. In addition, a standardized questionnaire was used for data collection, and majority of the questions were open-ended, thus reducing measurement bias.

This survey has some limitations that should be considered for a fair interpretation of the findings. First, the status of influenza vaccination was self-reported by the participants (there's

no yet an IV card in Tunisia); therefore, it could be subject to measurement bias due to poor recall, leading to underestimation of vaccine coverage. This source of bias should be minimal because women tend to memorize events related to pregnancy in our culture. Second, participants with a favorable attitude toward influenza vaccination may be more likely to respond to the questionnaire than those with a negative attitude, thereby introducing a selection bias. Third, our study sample included participants from public health centers alone (primary and secondary health care centers) and may not be representative of all PW in Tunisia. In fact, private and University hospitals were not included. This is especially true regarding PW attending private health settings, who may belong to a higher financial bracket than those attending public centers. Those attending university hospitals may have suffered more from pregnancy related complications. However, a study reported that the majority of the Tunisian population (two out of three) seeks healthcare from the public sector [37].

Fourth, our study was not designed to estimate vaccine coverage and further studies are needed to give a current estimation of this indicator among a largest sample size of PW in Tunisia.

Besides, our data showed that women have a high education level in a significant proportion.

Hence, this selection bias could lead to an overestimation of some of the positive factors associated with vaccination and that, despite this overestimation, our results show that an adequate communication and awareness strategy on vaccination with the high-risk populations for seasonal influenza.

Overall, despite these potential limitations, the main findings of this study advocate for urgent policies and programs to address obvious gaps in the use of IV, by a universally considered high priority group for this intervention in Tunisia.

## V. Conclusions

In this study, less than 40% of the surveyed PW were willing to be vaccinated during their next pregnancy and only 4.6% received the IV during their current pregnancy. Presence of comorbidities prior or during pregnancy was the main predictors of this vaccine uptake. Health care providers seem to be main "game changers" influencing maternal IV uptake. IV for PW should be integrated in the Maternal and Child Health preventive program in Tunisia.

In addition, further studies in private health care sector are also needed to better assess the IV acceptance and uptake among PW.

## Supporting information

**S1 Fig. Sampling method.**
(PDF)

**S1 Table. Distribution of the selected health care facilities' number by governorate.** Tunisia, 2018–2019.
(PDF)

**S2 Table. Distribution of the selected health care facilities' name by governorate, Tunisia, 2018–2019.**
(PDF)

**S3 Table. Factors associated with willingness to receive influenza vaccine during pregnancy (n = 999)[1].**
(PDF)

**S1 Appendix. Survey's questionnaire in French language.**
(PDF)

**S2 Appendix. Survey's questionnaire in English language.**
(PDF)

## Acknowledgments

We thank Margaret McCarron and Kathryn Lafond from the US CDC and Malembe Ebama from the Task Force for Global Health for contributing to the study design and the regional health directorates of Ariana, Ben Arous, Bizerte, Gafsa, Kairouan, Mahdia, Siliana and Sousse for participating to data collection.

## Author Contributions

**Conceptualization:** Sonia Dhaouadi, Leila Bouabid, Afif Ben Salah, Nissaf Bouafif ép Ben Alaya, Jihene Bettaieb.

**Data curation:** Sonia Dhaouadi, Amal Cherif, Ines Cherif, Hind Bouguerra, Leila Bouabid, Nourhene Najar, Nissaf Bouafif ép Ben Alaya, Jihene Bettaieb.

**Formal analysis:** Sonia Dhaouadi, Amal Cherif, Ines Cherif, Hind Bouguerra, Nourhene Najar, Nissaf Bouafif ép Ben Alaya, Jihene Bettaieb.

**Funding acquisition:** Afif Ben Salah.

**Investigation:** Sonia Dhaouadi, Ghassen Kharroubi, Leila Bouabid, Adel Gharbi, Nissaf Bouafif ép Ben Alaya, Jihene Bettaieb.

**Methodology:** Sonia Dhaouadi, Ghassen Kharroubi, Afif Ben Salah, Nissaf Bouafif ép Ben Alaya, Jihene Bettaieb.

**Project administration:** Leila Bouabid, Adel Gharbi, Nissaf Bouafif ép Ben Alaya, Jihene Bettaieb.

**Resources:** Leila Bouabid, Adel Gharbi, Afif Ben Salah, Nissaf Bouafif ép Ben Alaya, Jihene Bettaieb.

**Supervision:** Leila Bouabid, Afif Ben Salah, Nissaf Bouafif ép Ben Alaya, Jihene Bettaieb.

**Validation:** Afif Ben Salah, Nissaf Bouafif ép Ben Alaya, Jihene Bettaieb.

**Visualization:** Afif Ben Salah, Nissaf Bouafif ép Ben Alaya, Jihene Bettaieb.

**Writing – original draft:** Sonia Dhaouadi, Ghassen Kharroubi, Ines Cherif.

**Writing – review & editing:** Afif Ben Salah, Nissaf Bouafif ép Ben Alaya, Jihene Bettaieb.

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
