## [Decision Letter · Decision Letter 0]

18 Jun 2021

PONE-D-21-08415

Knowledge Attitudes and Practices toward seasonal influenza vaccine among pregnant women in Tunisia

PLOS ONE

Dear Dr. Bettaieb,

Thank you for submitting your manuscript to PLOS ONE. After careful consideration, we feel that it has merit but does not fully meet PLOS ONE’s publication criteria as it currently stands. Therefore, we invite you to submit a revised version of the manuscript that addresses the points raised during the review process.

This study is focusing on assessing the Knowledge Attitudes and Practices toward seasonal influenza vaccine among pregnant women in Tunisia which provides valuable support for the decision makers. However, revising the methodology section is very fundamental for accepting this article.  The concerns were mainly for the methodology section. 

Please note that your manuscript was reviewed by 5 experts in the field. There is consensus agreement that the idea of the article is interesting. Meanwhile, they have identified some important problems and provided copious comments. 

 The manuscript could be greatly strengthened by considering editing according to the specific attached Reviewers’ comments.

Please submit your revised manuscript by July 31 2021 11:59PM.  If you will need more time than this to complete your revisions, please reply to this message or contact the journal office at plosone@plos.org. Please include the following items when submitting your revised manuscript:

We look forward to receiving your revised manuscript.

Kind regards,

Ammal Mokhtar Metwally, Ph.D (MD)

Academic Editor

PLOS ONE

Journal Requirements:

2. Thank you for stating in the text of your manuscript "Ethical approval for this survey was obtained from the Biomedical Ethics Committee of the

Pasteur Institute of Tunis. Approval was also obtained from the Tunisian Ministry of Health. All women included in the survey were informed about the objectives and modalities of the study. Women who agreed to be interviewed provided written informed consent." Please also add this information to your ethics statement in the online submission form.

3. Please list the names of all of the public health centers where participants were recruited from.

4. Please include additional information regarding the survey or questionnaire used in the study and ensure that you have provided sufficient details that others could replicate the analyses. For instance, if you developed the survey or questionnaire as part of this study and it is not under a copyright more restrictive than CC-BY, please include a copy, in both the original language and English, as Supporting Information. If the questionnaire is published, please provide a citation to the (1) questionnaire and/or (2) original publication associated with the questionnaire.

Reviewers' comments:

Reviewer's Responses to Questions

**Comments to the Author**

1. Is the manuscript technically sound, and do the data support the conclusions?

Reviewer #1: Yes

Reviewer #2: Yes

Reviewer #3: Yes

Reviewer #4: Yes

Reviewer #5: Yes

2. Has the statistical analysis been performed appropriately and rigorously? 

Reviewer #1: Yes

Reviewer #2: Yes

Reviewer #3: Yes

Reviewer #4: Yes

Reviewer #5: I Don't Know

3. Have the authors made all data underlying the findings in their manuscript fully available?

Reviewer #1: Yes

Reviewer #2: Yes

Reviewer #3: Yes

Reviewer #4: Yes

Reviewer #5: Yes

4. Is the manuscript presented in an intelligible fashion and written in standard English?

Reviewer #1: Yes

Reviewer #2: Yes

Reviewer #3: Yes

Reviewer #4: Yes

Reviewer #5: Yes

5. Review Comments to the Author

Reviewer #1: 1. Sampling method - needs more clarity, because the author claims that the sample is a representative of the total population and also able generalize

2. How they arrive at Eight governorates out of 24 were randomly selected - Please clarify because the representation from all regions was not proportionate

3. This study included only qualitative data that were presented as numbers and percentages - Explain.

4. First, the status of influenza vaccination was self-reported by the participants – Self reporting could have been done with EVIDENCE CROSS CHECKED BY VACCINATION CARDS, MCH CARDS ETC. ???

Reviewer #2: 1. The Discussion section paragraph 1 and 2 are the repetitions of Aim and Results of the previous sections. I would consider it as redundant and may i suggest to remove it.

2. Suggested to reason out why the Flu shots uptake are high in U.S,UK and Ireland. This may give a lead to adopt it in Tunisia.

Suggested to correct as Strengths and Limitations and NOT as Strength and Limitations -Plural is required

Reviewer #3: This is an interesting manuscript about pregnant women's attitudes towards having an influenza vaccine in Tunisia. A major concern I have about the paper is that while information about women's attitudes to the vaccine have been sought, it appears that the main reason women do not get the vaccine is because it is not offered to them and it is not free of charge. However, I appreciate that the aim of the manuscript is to form an evidence base for the promotion of the vaccine to pregnant women. I think the manuscript could be enhanced by providing some context in the introduction about what vaccines pregnant women in Tunisia are/are not offered and any associated costs to the women. Even if the vaccine was offered access to getting the influenza vaccine could be a barrier to women, especially those who live rurally. This could also be touched on in the introduction.

Minor comments

Study design - Is March to May winter in Tunisia? Add in details about the season.

Study population - Is it normal for pregnant women to attend a public health center? Provide a figure on how many women usually attend these clinics.

Sample size - leave out the equation, just include the sentence about the sample size. This sentence could go with your study population information rather than under its own heading.

Data analysis - Move the sentences together to make two paragraphs.

Ethical considerations - "the" before data is not needed. "will be analyzed" should be "would be analyzed" "is voluntary" should be "was voluntary"

Results

First paragraph "Age" should be "age"

Discussion

You mention that the uptake of other low-income countries are similar to Tunisia. What are the reasons for this? Is it also because the vaccine is not offered/available? Does it cost money?

Reviewer #4: 1.Subject to qualitative data availability and usage, the authors have used two year old data for qualitative study.

2.Sampling among all the three regions are not appropriate, it can be elaborated to give a clear understanding. And what percentage of population has been used in sampling.

3.The study population targeted is all pregnant women more than or equal to 18 years, so less than 18 years of pregnant women can not be the part of sampling.

4.Elaborate the criteria on which the status of influenza vaccination was reported by the targeted population.

Reviewer #5: This paper presents the results of a national survey of attitudes toward influenza vaccination among pregnant women attending antenatal care in Tunisia. The authors report that the majority of women had heard of the vaccine, but only a small minority had been vaccinated, which was related to concerns about the safety and effectiveness of the vaccine. The authors conclude that health care professionals working in antenatal care can increase vaccination rates by encouraging their patients to get vaccinated.

The paper is well-written, with a well-justified aim and an interesting discussion. My main concern is that it is difficult to evaluate whether the results are generalizable to the study population (i.e., to all pregnant women attending antenatal care in the Tunisian public health care sector), as the sampling procedure is not described in detail and the response rate has not been reported. My specific comments can be found below.

Abstract/Introduction:

1. Should vaccine uptake be included in the aims?

2. In reference 16, I cannot find the WHO:s target of 80% of pregnant women being vaccinated.

Methods:

3. The authors write that sampling was stratified by region, but how was the number of governorates sampled from each region determined?

4. What was the second sampling stage? Were hospitals/health care centers sampled within governorates? According to the text, after governorates had been sampled, “stratified sampling was conducted based on the selected governorates”. Could the authors please clarify what was sampled and what the stratification variable(s) was (were)?

5. How were hospitals/health care centers within governorates sampled?

6. Were all women attending antenatal care at the sampled hospitals/health care centers asked to participate, or was a sample of women taken?

7. The authors write that the sample was self-weighted and that “[t]he sample size was distributed according to the distribution of the target population in the different regions.” Does this mean that the number of sampled patients in each governorate was proportional to the size of the total population size?

8. Was the sampling design taken into account in the data analysis? If not, please explain why.

9. The authors explain that the interviewers were trained. However, it is unclear who the interviewers were. Were they, for example, nurses or other health care professionals already working at the hospitals/health care centers, or were they external staff?

10. In the Data Analysis section, I think “qualitative data” should say “qualitative variable”, as the data are quantitative (i.e., numbers), but the variables are qualitative (i.e., ordinal/nominal).

11. In the Data Analysis section, it would be good to mention that odds ratios and 95% confidence intervals were calculated.

Results:

12. How many health care centers/hospitals were sampled?

13. How many health care centers/hospitals were asked to participate in the survey? How many were willing to participate?

14. How many women were assessed for eligibility? How many were considered eligible? How many were excluded? What was the response rate in the survey?

15. In the second paragraph, should the word “nearly” be replaced with “more than”?

16. In the third paragraph, please include not only the percentage, but also the number, of women who received the vaccine in the past or during their current pregnancy.

17. In the third paragraph, should “FIV” be “FV”?

18. In the sixth paragraph, should the word “except” be deleted or replaced with another word?

19. In the last paragraph of the Results section, the authors mention two reasons why women refused getting vaccinated (fear of side effects and concerns regarding self-harm/harm to the fetus). Was this question open-ended? If it was, did the authors consider categorizing all responses, instead of just two? If it was not open-ended, what were the possible responses? The answers “fear of side effects” and “concerns regarding self-harm/harm to the fetus” seem identical to me. Could these answers be combined?

20. In the last paragraph of the Results section, it says that the main reason that patients were not willing get vaccinated was concerns about adverse effects. However, in Table 2, it looks like vaccinated and non-vaccinated women differ more in their beliefs about vaccine effectiveness than vaccine safety. Could this result mean that concerns about lack of effectiveness are an even more important reason for the women’s unwillingness to get vaccinated than concerns about adverse effects?

Discussion/Conclusion:

21. Would it be interesting to mention that only 48% of women claim that they would take the vaccine if it were recommended by a health care professional and provided free of charge, which is substantially lower than the target of 80% vaccinated among pregnant women, even if this would be a large improvement from the current 4%? In addition, would it be interesting to mention that as many as 75% say they would be willing to take it if it was recommended by a doctor, but only 34% if it was recommended by another health care professional?

22. Page 8, Paragraph 3: Is “avail” the right word here?

23. Page 8, last paragraph: Most questions do not seem to be open-ended, as they are multiple-choice.

Table 2:

24. Is there a round-off error in the odds ratio for the first question (“Do you know anyone who has been severely ill with influenza?”)? I get an odds ratio of 0.95 (=136*360/272*189), which rounds up to 1.0.

References

25. Parts of the references are in French.

6. PLOS authors have the option to publish the peer review history of their article (what does this mean?). If published, this will include your full peer review and any attached files.

Reviewer #1: No

Reviewer #2: No

Reviewer #3: No

Reviewer #4: **Yes: **Rupam Bharti

Reviewer #5: **Yes: **Jonathan Bergman

---

## [Author Response · Author response to Decision Letter 0]

30 Aug 2021

Rebuttal letter

 Tunis, August 29, 2021

Plos One Journal

Dear editors and reviewers

I am pleased, on behalf of all co-authors, to submit to your consideration the revised version of our manuscript newly titled “Knowledge Attitudes and Practices toward seasonal influenza vaccine among pregnant women during the 2018/2019 influenza season in Tunisia” [Submission ID: [PONE-D-21-08415] - [EMID:8bb08bb7b6d30319] by Sonia Dhaouadi and al for publication in Plos One. 

We are grateful to the editor and reviewers for taking time to read the manuscript and for their valuable feedback and comments that we have taken into account.

We respond hereafter point by point to all comments raised. All changes and corrections are highlighted in green in the revised version of the manuscript.

I hope that revisions have improved the quality and the clarity of the paper and that it is now suitable for publication in Plos One.

Sincerely,

Sonia Dhaouadi

National Observatory of New and Emerging Diseases, Tunis, Tunisia

sonidhaouadi88@gmail.com

Authors’ response: Done

2. Thank you for stating in the text of your manuscript "Ethical approval for this survey was obtained from the Biomedical Ethics Committee of the Pasteur Institute of Tunis. Approval was also obtained from the Tunisian Ministry of Health. All women included in the survey were informed about the objectives and modalities of the study. Women who agreed to be interviewed provided written informed consent." Please also add this information to your ethics statement in the online submission form.

Authors’ response: Done. 

3. Please list the names of all of the public health centers where participants were recruited from.

Authors’ response: This information was added in Appendix S2.

4. Please include additional information regarding the survey or questionnaire used in the study and ensure that you have provided sufficient details that others could replicate the analyses. For instance, if you developed the survey or questionnaire as part of this study and it is not under a copyright more restrictive than CC-BY, please include a copy, in both the original language and English, as Supporting Information. If the questionnaire is published, please provide a citation to the (1) questionnaire and/or (2) original publication associated with the questionnaire.

Authors’ response: Questionnaire survey was added in appendix 3 in both French and English languages (Appendix 3 a-b).

Authors’ response: Done (S1 Table, S1 Appendix, S2 Appendix, S3-a Appendix and S3-b Appendix)

Please review your reference list to ensure that it is complete and correct. If you have cited papers that have been retracted, please include the rationale for doing so in the manuscript text, or remove these references and replace them with relevant current references. Any changes to the reference list should be mentioned in the rebuttal letter that accompanies your revised manuscript. If you need to cite a retracted article, indicate the article’s retracted status in the References list and also include a citation and full reference for the retraction -notice.

Authors’ response:

References added:16, 17, 18,20,21, 22,25,26 and 27.

Reviewer #1: 1. Sampling method - needs more clarity, because the author claims that the sample is a representative of the total population and also able generalize

Authors’ response: We conducted a multiple stage sampling according to region, governorate and area of residence. More details are show in Methods section (paragraph 3: sampling procedure, line 112-line 126) and in appendix S1.

2. How they arrive at Eight governorates out of 24 were randomly selected - Please clarify because the representation from all regions was not proportionate

Authors’ response: Given the limited budget of the survey (limited logistics support and resources), we choose to select the one third of the total 24 Tunisian governorates (24/3=8). Then eight governorates were randomly selected.

3. This study included only qualitative data that were presented as numbers and percentages - Explain.

Authors’ response: In order to facilitate the analysis, we decided to categorize them into qualitative variables.

4. First, the status of influenza vaccination was self-reported by the participants – Self reporting could have been done with EVIDENCE CROSS CHECKED BY VACCINATION CARDS, MCH CARDS ETC. ???

Authors’ response: There’s not yet an influenza vaccination card in Tunisia (Discussion, line 340-line 341)

Reviewer #2: 1. The Discussion section paragraph 1 and 2 are the repetitions of Aim and Results of the previous sections. I would consider it as redundant and may i suggest to remove it.

Authors’ response: We took into account this comment.

2. Suggested to reason out why the Flu shots uptake are high in U.S,UK and Ireland. This may give a lead to adopt it in Tunisia.

Authors’ response: This comment was taken into consideration in Discussion part (line 263-line 277) 

In those countries (USA and OK), PW are listed and invited to receive the IV. In addition, the healthcare provider can recommend, offer, and administer the vaccine at the same visit as opposed to making a recommendation and referring the patient elsewhere to receive the vaccine. There’s a close collaboration between medical doctors and midwives to ensure the enhance the coverage rate.

Suggested to correct as Strengths and Limitations and NOT as Strength and Limitations -Plural is required

Authors’ response: Done

Reviewer #3: This is an interesting manuscript about pregnant women's attitudes towards having an influenza vaccine in Tunisia. A major concern I have about the paper is that while information about women's attitudes to the vaccine have been sought, it appears that the main reason women do not get the vaccine is because it is not offered to them and it is not free of charge. However, I appreciate that the aim of the manuscript is to form an evidence base for the promotion of the vaccine to pregnant women. I think the manuscript could be enhanced by providing some context in the introduction about what vaccines pregnant women in Tunisia are/are not offered and any associated costs to the women. Even if the vaccine was offered access to getting the influenza vaccine could be a barrier to women, especially those who live rurally. This could also be touched on in the introduction.

Authors’ response: This idea was taken into account in the introduction. The IV is not free of charge and non-refundable among PW in Tunisia.

Minor comments

Study design - Is March to May winter in Tunisia? Add in details about the season.

Authors’ response: The 2018/2019 influenza season in Tunisia from 1st October 2018 to end of April 2019. The survey’s period (March-May 2019) coincided with spring season in Tunisia.

Study population - Is it normal for pregnant women to attend a public health center? Provide a figure on how many women usually attend these clinics.

Authors’ response: Given the high medical fees of consultation in private sector, PW in Tunisia attended either public and private sector. However, we don’t have exactly the number of PW attending the public health care facilities in public sector. 

Sample size - leave out the equation, just include the sentence about the sample size. This sentence could go with your study population information rather than under its own heading.

Authors’ response: Done (the sample size paragraph was eliminated).

Data analysis - Move the sentences together to make two paragraphs.

Authors’ response: Done

Ethical considerations - "the" before data is not needed. "will be analyzed" should be "would be analyzed" "is voluntary" should be "was voluntary"

Authors’ response: Done

Results

First paragraph "Age" should be "age"

Authors’ response: Done

Discussion

You mention that the uptake of other low-income countries are similar to Tunisia. What are the reasons for this? Is it also because the vaccine is not offered/available? Does it cost money?

Authors’ response: In Low and Middle income countries, the low IV coverage could be explained by the difficulty of effective implementation of National Strategic Plan for Vaccines among high risk groups, and by the lack of awareness of burden of disease by decisions-makers and by the target population. In Tunisia, it’s more related to the influenza vaccine awareness among pregnant women and Health care workers than the vaccine supplies and health care access. The vaccine is not free of charge and non-refundable among this group.

Reviewer #4: 1. Subject to qualitative data availability and usage, the authors have used two-year-old data for qualitative study.

Authors’ response: We conducted a quantitative study.

2.Sampling among all the three regions are not appropriate, it can be elaborated to give a clear understanding. And what percentage of population has been used in sampling.

Authors’ response: Please see response to comment 1 reviewer 1.

3.The study population targeted is all pregnant women more than or equal to 18 years, so less than 18 years of pregnant women cannot be the part of sampling.

Authors’ response: For ethical reasons (PW aged less than 18 years must to give the written consent of her legal tutor (her parent or her husband) before participation in the survey. So we decided to include only PW aged 18 years and above.

Indeed, we used the published data for the National Institute of Statistics (INS) about the 2014 population census. The repartition of number of women of childbearing age (15-49 years) by age group was as follows (five-year interval): 15-19, 20-24,25-29,30-34,35-39,40-44 and 45-49 years. We can’t use the age >=18 for the sampling.

4.Elaborate the criteria on which the status of influenza vaccination was reported by the targeted population.

Authors’ response: We took into consideration this comment in method section (line 134-136)

Reviewer #5: This paper presents the results of a national survey of attitudes toward influenza vaccination among pregnant women attending antenatal care in Tunisia. The authors report that the majority of women had heard of the vaccine, but only a small minority had been vaccinated, which was related to concerns about the safety and effectiveness of the vaccine. The authors conclude that health care professionals working in antenatal care can increase vaccination rates by encouraging their patients to get vaccinated.

The paper is well-written, with a well-justified aim and an interesting discussion. My main concern is that it is difficult to evaluate whether the results are generalizable to the study population (i.e., to all pregnant women attending antenatal care in the Tunisian public health care sector), as the sampling procedure is not described in detail and the response rate has not been reported. My specific comments can be found below.

Abstract/Introduction:

1. Should vaccine uptake be included in the aims?

Authors’ response: Yes, assess influenza vaccine uptake was added in the aims au suggested. (Abstract, Background part, line 38).

2. In reference 16, I cannot find the WHO’s target of 80% of pregnant women being vaccinated.

Authors’ response: Reference 16 was updated as mentioned.

Methods:

3. The authors write that sampling was stratified by region, but how was the number of governorates sampled from each region determined?

Authors’ response: We applied the percentage of Women of childbearing age in each region (North, Center and South) to determine the number of selected governorates in each region: 

48.03% (North)*8=4 governorates.

36.72% (Center)*8=3 governorates

15.25% (South)*8=1 governorate

The governorates were then randomly selected in each region.

More details are shown in appendix 1.

4. What was the second sampling stage? Were hospitals/health care centers sampled within governorates? According to the text, after governorates had been sampled, “stratified sampling was conducted based on the selected governorates”. Could the authors please clarify what was sampled and what the stratification variable(s) was (were)?

Authors’ response: We conducted a multiple stage sampling according to 3 stratified variables: region, governorate and area of residence. More details are show in Methods section (paragraph 3: sampling procedure, line 112-line 126) and in appendix S1.

5. How were hospitals/health care centers within governorates sampled?

Authors’ response: All public health care centers (primary and secondary health care centers) with antenatal care service were included (Methods, line 124-126)

6. Were all women attending antenatal care at the sampled hospitals/health care centers asked to participate, or was a sample of women taken?

Authors’ response: Yes, all women attending antenatal care at the sampled hospitals/health care centers at the time of the survey were asked to participate.

7. The authors write that the sample was self-weighted and that “the sample size was distributed according to the distribution of the target population in the different regions.” Does this mean that the number of sampled patients in each governorate was proportional to the size of the total population size?

Authors’ response: Yes; that the number of sampled patients in each governorate was proportional to the size of the women of childbearing age’s population size.

.

8. Was the sampling design taken into account in the data analysis? If not, please explain why.

Authors’ response: No, we did a self-weighted sampling taking into account the distribution of the woman of childbearing age according to region governorates and area of residence as we don't have such information for pregnant women.

9. The authors explain that the interviewers were trained. However, it is unclear who the interviewers were. Were they, for example, nurses or other health care professionals already working at the hospitals/health care centers, or were they external staff?

Authors’ response: The interviewers were health care worker in public health sector (Methods, line 130-131).

10. In the Data Analysis section, I think “qualitative data” should say “qualitative variable”, as the data are quantitative (i.e., numbers), but the variables are qualitative (i.e., ordinal/nominal).

Authors’ response: Done

11. In the Data Analysis section, it would be good to mention that odds ratios and 95% confidence intervals were calculated. 

Authors’ response: This comment was taken into consideration in Methods part (line 154-155).

Results:

12. How many health care centers/hospitals were sampled?

Authors’ response: 84

13. How many health care centers/hospitals were asked to participate in the survey? How many were willing to participate?

Authors’ response: The sampled 84 health care centers/hospitals were asked to participate in the survey and were willing to participate.

14. How many women were assessed for eligibility? How many were considered eligible? How many were excluded? What was the response rate in the survey?

Authors’ response: The number of pregnant women who were approached to participate in the survey as well as the response rate were added in the first paragraph of result part (line 169-line 172).

15. In the second paragraph, should the word “nearly” be replaced with “more than”?

Authors’ response: Nearly three quarter (74.7%) was replaced by more than half as suggested (line 173 in Results section).

16. In the third paragraph, please include not only the percentage, but also the number, of women who received the vaccine in the past or during their current pregnancy.

Authors’ response: The number of women who received the vaccine at least once in the past and during the current pregnancy was added (line 183-184).

17. In the third paragraph, should “FIV” be “FV”?

Authors’ response: Flu vaccination (FV) was replaced by influenza vaccination (IV) in all the manuscript.

18. In the sixth paragraph, should the word “except” be deleted or replaced with another word?

Authors’ response: except was replaced by both of women (Results, line 192).

19. In the last paragraph of the Results section, the authors mention two reasons why women refused getting vaccinated (fear of side effects and concerns regarding self-harm/harm to the fetus). Was this question open-ended? If it was, did the authors consider categorizing all responses, instead of just two? If it was not open-ended, what were the possible responses? The answers “fear of side effects” and “concerns regarding self-harm/harm to the fetus” seem identical to me. Could these answers be combined?

Authors’ response:

-Question related to the main reasons of IV refuse was an open-ended question (question 30 in the questionnaire). There were three main reasons of IV refuse:

* Fear of side effects” and “concerns regarding self-harm/harm to the fetus (72,1%)

* Concerns about vaccine efficacy (16.2%)

* Natural immunity inducted by the infection is better than the immunity inducted by the vaccine (11.2%) (Results section line 214-217).

20. In the last paragraph of the Results section, it says that the main reason that patients were not willing get vaccinated was concerns about adverse effects. However, in Table 2, it looks like vaccinated and non-vaccinated women differ more in their beliefs about vaccine effectiveness than vaccine safety. Could this result mean that concerns about lack of effectiveness are an even more important reason for the women’s unwillingness to get vaccinated than concerns about adverse effects?

Authors’ response: The question related to the main reasons to refuse IV was an open ended question (question 31 in the questionnaire). Questions related to the vaccine efficacy/vaccine safety presented in Table 3 was a multiple choice question (question 26 in the questionnaire, based on five points Likert scale). The response of interviewed women differed regarding the priority according to the type of question. It seems that vaccine safety concerns were more important than vaccine efficacy concerns when it was a spontaneous response (open ended question), unlike to multiple choice question (inducted response) where vaccine efficacy concerns were more priority.

Discussion/Conclusion:

21. Would it be interesting to mention that only 48% of women claim that they would take the vaccine if it were recommended by a health care professional and provided free of charge, which is substantially lower than the target of 80% vaccinated among pregnant women, even if this would be a large improvement from the current 4%? In addition, would it be interesting to mention that as many as 75% say they would be willing to take it if it was recommended by a doctor, but only 34% if it was recommended by another health care professional?

Authors’ response: This idea was taken into account in discussion section (line 284-290).

22. Page 8, Paragraph 3: Is “avail” the right word here?

Authors’ response: Avail was replaced by uptake as suggested (Discussion line 311).

23. Page 8, last paragraph: Most questions do not seem to be open-ended, as they are multiple-choice.

Authors’ response: Yes, it was a Likert scale question based on 5-point agreement scale (Table 3).

Table 2:

24. Is there a round-off error in the odds ratio for the first question (“Do you know anyone who has been severely ill with influenza?”)? I get an odds ratio of 0.95 (=136*360/272*189), which rounds up to 1.0.

Authors’ response:

OR was rounded to 1.0 in Table 2 as suggested.

References

25. Parts of the references are in French.

Authors’ response:

The English title was added in those references (Ref 9 and 40).

Thank you again for taking the time to share your insightful feedback.

---

## [Decision Letter · Decision Letter 1]

15 Oct 2021

PONE-D-21-08415R1Knowledge Attitudes and Practices toward seasonal influenza vaccine among pregnant women during the 2018/2019 influenza season in TunisiaPLOS ONE

Dear Dr. Bettaieb,

Thank you for submitting your manuscript to PLOS ONE. After careful consideration, we feel that it has merit but does not fully meet PLOS ONE’s publication criteria as it currently stands. Therefore, we invite you to submit a revised version of the manuscript that addresses the points raised during the review process.

Great effort was made by the authors to utilize the feedback that was provided for them to correct for resubmission. I find it interesting and improved with respect to the original submission. There are still some issues to be clarified and things to adjust mainly for the methodology section to achieve the aim (enclosed). 

Please note the following:

You are invited to publish the peer review history of your articles. Please note that, if this manuscript is accepted for publication and you and your authors opt-in to publishing the peer review history, all decision letters - which may include all comments will be published, along with your responses to reviewer comments.

Please also note that PLOS ONE is currently running a Call for Papers on the theme of Influenza Prevention (https://collections.plos.org/call-for-papers/influenza/).

While during your submission, you did not specifically submit to this Call for Papers, we feel that the manuscript falls within its scope. If you and your co-authors would like your manuscript to be considered by the Guest Editors for inclusion in the Influenza Collection or if you have any questions about this matter, or on this Call for Papers more generally, do not hesitate to email shepp@plos.org.

We look forward to receiving your revised manuscript.

Kind regards,

Ammal Mokhtar Metwally, Ph.D (MD)

Academic Editor

PLOS ONE

Journal Requirements:

Reviewers' comments:

Reviewer's Responses to Questions

**Comments to the Author**

1. If the authors have adequately addressed your comments raised in a previous round of review and you feel that this manuscript is now acceptable for publication, you may indicate that here to bypass the “Comments to the Author” section, enter your conflict of interest statement in the “Confidential to Editor” section, and submit your "Accept" recommendation.

Reviewer #1: All comments have been addressed

Reviewer #2: All comments have been addressed

Reviewer #5: (No Response)

2. Is the manuscript technically sound, and do the data support the conclusions?

Reviewer #1: Yes

Reviewer #2: Yes

Reviewer #5: Partly

3. Has the statistical analysis been performed appropriately and rigorously? 

Reviewer #1: Yes

Reviewer #2: Yes

Reviewer #5: I Don't Know

4. Have the authors made all data underlying the findings in their manuscript fully available?

Reviewer #1: Yes

Reviewer #2: (No Response)

Reviewer #5: Yes

5. Is the manuscript presented in an intelligible fashion and written in standard English?

Reviewer #1: Yes

Reviewer #2: Yes

Reviewer #5: Yes

6. Review Comments to the Author

Reviewer #1: Congrats to the authors on the KAP study on Infleunza vaccine and the same can be attempted towards COVID 19 vaccines

Reviewer #2: The authors have addressed all the comments and suggestions. Look forward to similar works in the future.

Reviewer #5: The authors have improved their paper, but I still have a few questions and comments. Some of the comments concern minor unclarities or inconsistencies in the data that I did not notice the first time, which I apologize for. Other comments concern the description of the sampling procedure, which has been improved but can still be clarified further.

Major issues:

1. Methods, Line 113: How did the authors determine that 8 governorates should be sampled? In their rebuttal, they explain that this was determined by budget constraints, as 8 corresponds to 1/3 of Tunisia’s 24 governorates. Pease add this information to the manuscript.

2. Methods, Lines 112-113: It could be clarified how the number of governorates sampled per region was determined. I know this information is provided in Lines 120-123, but the authors provide a much clearer explanation in their rebuttal (i.e. 8 * % of Tunisia’s women of childbearing age living in a region). I recommend that the authors be as clear in the manuscript as they are in their rebuttal.

3. Methods, 118-119: It is unclear how area of residence (urban/rural) was relevant to the sampling procedure. According to the authors’ rebuttal, all public health centers in the sampled governorates were included. If so, shouldn’t area of residence be irrelevant?

4. Methods, Sampling procedure: In the authors’ rebuttal, it sounds like they performed region-stratified, cluster sampling, meaning that they first randomly sampled governorates from each region and then included all pregnant women attending any public health center in the sampled governorates. If so, this could be clarified in the Sampling Procedure section.

5. Results, Line 169: The number of survey respondents was 1200, which is identical to the calculated sample size reported in the Methods section. This seems unlikely if the authors really did recruit all pregnant women attending any public health center during a 3-month period. Are the authors’ sure they didn't recruit consecutive patients and then stop when the calculated sample had been reached?

6. S1 Appendix: The sampling method illustrated here looks like it is stratified sampling by region, governorate, and area of residence. As mentioned above, however, only region seems to have been a stratification variable.

7. Rebuttal: The authors explain that that they did not take the sampling procedure into account during the analysis due to a lack of information to do so. However, ignoring cluster sampling in an analysis can lead to confidence intervals that are too narrow. Have the authors considered this possibility?

Minor issues:

8. Abstract, Line 48: The lower bound of the confidence interval is 57.7% here, but it is 57.6% in the main text (Results, Line 182). Please check this.

9. Abstract, Lines 48-49: The text states that 4.4% of women who were aware of the vaccine received it during their current pregnancy; however, this percentage seems to be calculated based on all 1157 included women (51/1157=4.4%). The same was done in Results, Lines 182-184, both for women who had received the vaccine during the current pregnancy and for those who had received it sometime in the past. Please check this.

10. Results, Line 163: Could the ID numbers be linked to personal identifiers? If so, the data would be coded/pseudo-anonymized rather than anonymized.

11. Results, Lines 181-182: Is there a round-off error in the percentage of women who had heard of influenza, 85.5%? I get 988/1157=84.4%. Is there an error in the calculation of women who had heard of the vaccine, 60.3%? I get 694/1157=60.0%.

12. Results, Line 187: Consider adding the number of women who would be willing to receive the vaccine, instead of just reporting the percentage.

13. Discussion, Line 307: How was 6.5% calculated (numerator/denominator)? According to Table 2, 72 patients reported having enough information about side effects. Is 72 the numerator used?

14. References: There are still French words here.

7. PLOS authors have the option to publish the peer review history of their article (what does this mean?). If published, this will include your full peer review and any attached files.

Reviewer #1: No

Reviewer #2: No

Reviewer #5: **Yes: **Jonathan Bergman

---

## [Author Response · Author response to Decision Letter 1]

26 Nov 2021

Rebuttal letter

 Tunis, November 26, 2021

Plos One Journal

Dear editors and reviewers

I am pleased, on behalf of all co-authors, to submit to your consideration the2nd revised version of our manuscript newly titled “Knowledge Attitudes and Practices toward seasonal influenza vaccine among pregnant women during the 2018/2019 influenza season in Tunisia” [Submission ID: [PONE-D-21-08415] - [EMID:8bb08bb7b6d30319] by Sonia Dhaouadi et al for publication in Plos One. 

We are grateful to the editor and reviewers for taking time to read the manuscript and for their valuable feedback and comments that we have taken into account.

We respond hereafter point by point to all comments raised. All changes and corrections are highlighted in green in the revised version of the manuscript.

I hope that revisions have improved the quality and the clarity of the paper and that it is now suitable for publication in Plos One.

Sincerely,

Sonia Dhaouadi

National Observatory of New and Emerging Diseases, Tunis, Tunisia

sonidhaouadi88@gmail.com

00216 95334250

Major issues:

1. Methods, Line 113: How did the authors determine that 8 governorates should be sampled? In their rebuttal, they explain that this was determined by budget constraints, as 8 corresponds to 1/3 of Tunisia’s 24 governorates. Pease add this information to the manuscript.

Authors’ response: We took into consideration your suggestion and we added this information to the manuscript (Methods, lines 116-117).

2. Methods, Lines 112-113: It could be clarified how the number of governorates sampled per region was determined. I know this information is provided in Lines 120-123, but the authors provide a much clearer explanation in their rebuttal (i.e. 8 * % of Tunisia’s women of childbearing age living in a region). I recommend that the authors be as clear in the manuscript as they are in their rebuttal.

Authors’ response: Your suggestion was taken into account. Please see Methods section, lines 117-123.

3. Methods, 118-119: It is unclear how area of residence (urban/rural) was relevant to the sampling procedure. According to the authors’ rebuttal, all public health centers in the sampled governorates were included. If so, shouldn’t area of residence be irrelevant?

Authors’ response: We apologize; it seems that we made an error when explaining the sampling method in the rebuttal letter. We meant that all public health centers were targeted by the selection procedure. Indeed, as we explained in the response to comment 4, not all public health centers were selected. In addition, we took into consideration area of residence in the self-weighting process in order to enhance similarity of distribution between our study population and the target population. It should be noted that area of residence refers to the residence of PW and not to the location of the health care facility. Some PW living in rural area may visit health care centers in urban area and vise-versa.

4. Methods, Sampling procedure: In the authors’ rebuttal, it sounds like they performed region-stratified, cluster sampling, meaning that they first randomly sampled governorates from each region and then included all pregnant women attending any public health center in the sampled governorates. If so, this could be clarified in the Sampling Procedure section.

Authors’ response: We reiterate our apology for the error when explaining the sampling method in the rebuttal letter. In fact, we performed a stratification according to region. Governorates were randomly selected (four in the North, three in the Center and one in the South). However, we did not proceed to a cluster sampling. Indeed, in each governorate we randomly selected health care facilities among those offering antenatal care visits as the survey progressed. Every day, we randomly selected one center among all those that offer medical care consultations for PW on that day. The selected center will not be selected again. 

On the day of their visit, investigator(s) were asked to randomly select participants from the exhaustive list of PW consulting the selected health care facility on that day. The number of PW approached to participate to the survey in each health care facility depended on the availability of human and logistic resources. 

It should be noted that a pilot study was performed in order to assess the comprehensiveness of questionnaire items, to identify ambiguous questions and to estimate the maximum number of questionnaires that could be properly administrate by one investigator which was equal to 20. The sampling procedure was clarified as suggested in sampling procedure (Please see lines 114-136).

5. Results, Line 169: The number of survey respondents was 1200, which is identical to the calculated sample size reported in the Methods section. This seems unlikely if the authors really did recruit all pregnant women attending any public health center during a 3-month period. Are the authors’ sure they didn't recruit consecutive patients and then stop when the calculated sample had been reached?

Authors’ response: Thank you for your comment. In fact, you are right we stopped recruitment of participants when the needed sample size was achieved. 

6. S1 Appendix: The sampling method illustrated here looks like it is stratified sampling by region, governorate, and area of residence. As mentioned above, however, only region seems to have been a stratification variable.

Authors’ response: We totally agree with you; region was the only stratification variable.

Details regarding sampling procedure were clarified on Methods section.

7. Rebuttal: The authors explain that they did not take the sampling procedure into account during the analysis due to a lack of information to do so. However, ignoring cluster sampling in an analysis can lead to confidence intervals that are too narrow. Have the authors considered this possibility?

Authors’ response: We totally agree with you. Not considering cluster sampling in analysis can lead to narrow confidence intervals. However, in the present study we did not proceed to a cluster sampling. In fact, we randomly selected daily health care facilities among those offering antenatal care visits as the survey progressed. On the day of their visit, investigator(s) were asked to randomly select participants from the exhaustive list of PW consulting the selected health care facility on that day. The number of PW approached to participate to the survey in each health care facility depended on the availability of human and logistic resources. The details were shown in sampling procedure (Methods). Please see lines (114-136).

Minor issues:

8. Abstract, Line 48: The lower bound of the confidence interval is 57.7% here, but it is 57.6% in the main text (Results, Line 182). Please check this.

Authors’ response: Thank you for your comment. The lower bound of the confidence interval is in fact 57.6%. We corrected the mistake in the abstract (Please see line 48).

9. Abstract, Lines 48-49: The text states that 4.4% of women who were aware of the vaccine received it during their current pregnancy; however, this percentage seems to be calculated based on all 1157 included women (51/1157=4.4%). The same was done in Results, Lines 182-184, both for women who had received the vaccine during the current pregnancy and for those who had received it sometime in the past. Please check this.

Authors’ response: Your suggestion was taken into consideration (Please see line 49 in Abstract and line 199 in Manuscript).

10. Results, Line 163: Could the ID numbers be linked to personal identifiers? If so, the data would be coded/pseudo-anonymized rather than anonymized.

Authors’ response: Data were coded anonymized and the ID number was not linked to personal identifiers.

11. Results, Lines 181-182: Is there a round-off error in the percentage of women who had heard of influenza, 85.5%? I get 988/1157=84.4%. Is there an error in the calculation of women who had heard of the vaccine, 60.3%? I get 694/1157=60.0%.

Authors’ response: In calculation, we only took into account valid percentage.

Women who had heard of influenza: n=988; missing data=2; 

The valid percentage=988/1155=85.5%; 95% CI [83.4%–87.5%].

Women who had heard of influenza vaccine: 

n=694; missing data=7; The valid percentage=694/1150=60.3%; 95% CI [57.5%–63.2%].

12. Results, Line 187: Consider adding the number of women who would be willing to receive the vaccine, instead of just reporting the percentage.

Authors’ response: This suggestion was taken into consideration: Results, line 207. (Yes=421, No=578, I don’t know=154, missing data=4; the valid percentage=421/1153=36.5%)

13. Discussion, Line 307: How was 6.5% calculated (numerator/denominator)? According to Table 2, 72 patients reported having enough information about side effects. Is 72 the numerator used? 

Authors’ response: The calculated percentage regarding information about side effects of IV: 6.5% (valid percentage) =75/1146 (Yes=75; No=1015; I don’t know=56; missing data=11). This information was added in Results section: lines 196-197.

Concerning table 2, we took into account only PW who answered Yes or No about willing to receive IV in the next pregnancy, participants who answered I don’t know about willingness to receive IV were excluded (Please see footnote Table 2). Participants who answered I don’t Know about having enough information about side effects were also excluded. Among the 75 PW who reported they had enough information about IV side effects, two answered I don’t know when asked about willingness to receive IV in the next pregnancy, so they were not included in Table 2.

14. References: There are still French words here.

Authors’ response: All references were corrected into English language.

Response to Editor Dr. Deshayne: 

We reported 95% confidence interval of crude odds ratio in text (Results section) to be more informative as suggested (lines 211, 212, 216, 218 and 219)

Thank you again for taking the time to share your insightful feedback and for taken into consideration our request that our manuscript be considered for inclusion in your ongoing Influenza collection (https://collections.plos.org/collection/influenza/)

---

## [Decision Letter · Decision Letter 2]

17 Dec 2021

PONE-D-21-08415R2Knowledge Attitudes and Practices toward seasonal influenza vaccine among pregnant women during the 2018/2019 influenza season in TunisiaPLOS ONE

Dear Dr. Bettaieb,

Thank you for submitting your manuscript to PLOS ONE. After careful consideration, we feel that it has merit but does not fully meet PLOS ONE’s publication criteria as it currently stands. Therefore, we invite you to submit a revised version of the manuscript that addresses the points raised during the review process.

We look forward to receiving your revised manuscript.

Kind regards,

Ammal Mokhtar Metwally, Ph.D (MD)

Academic Editor

PLOS ONE

Journal Requirements:

Reviewers' comments:

Reviewer's Responses to Questions

**Comments to the Author**

1. If the authors have adequately addressed your comments raised in a previous round of review and you feel that this manuscript is now acceptable for publication, you may indicate that here to bypass the “Comments to the Author” section, enter your conflict of interest statement in the “Confidential to Editor” section, and submit your "Accept" recommendation.

Reviewer #5: (No Response)

2. Is the manuscript technically sound, and do the data support the conclusions?

Reviewer #5: Yes

3. Has the statistical analysis been performed appropriately and rigorously? 

Reviewer #5: I Don't Know

4. Have the authors made all data underlying the findings in their manuscript fully available?

Reviewer #5: Yes

5. Is the manuscript presented in an intelligible fashion and written in standard English?

Reviewer #5: Yes

6. Review Comments to the Author

Reviewer #5: I thank the authors for further clarifying their paper. I have only three minor comments left.

Minor comments:

1. Page 4: In their rebuttal letter, the authors confirm my previous suspicion that the only stratification variable was region. However, now that the authors have further clarified their work, I think that the sampling procedure was also stratified by area of residence, as healthcare centers were selected from both urban and rural areas.

2. Page 4, Lines 126-127: The authors explain that patients were randomly selected at healthcare centers, but the authors may also want to explain how this was done in practice. For example, did the study staff enter the total number of patients on the patient list into a computer, which then returned a certain number of randomly selected patients, such as patients 2, 5, and 6 on the list?

3. Page 5: The authors have not addressed my question of whether it is appropriate to ignore the sampling procedure in the analysis. If I am not mistaken, Appendix S1 contains all the information required to incorporate this information. (The percentages at each stage of the sampling procedure could be used to construct sampling weights, could they not?)

7. PLOS authors have the option to publish the peer review history of their article (what does this mean?). If published, this will include your full peer review and any attached files.

Reviewer #5: **Yes: **Jonathan Bergman

---

## [Author Response · Author response to Decision Letter 2]

15 Feb 2022

Plos One Journal

Dear editors and reviewers,

I am pleased, on behalf of all co-authors, to submit to your consideration the revised version of our manuscript newly titled “Knowledge Attitudes and Practices toward seasonal influenza vaccine among pregnant women during the 2018/2019 influenza season in Tunisia” [Submission ID: [PONE-D-21-08415] - [EMID:8bb08bb7b6d30319] by Sonia Dhaouadi and al for publication in Plos One. 

We are grateful to the editor and reviewers for taking time to read the manuscript and for their valuable feedback and comments that we have taken into account.

We respond hereafter point by point to all comments raised. All changes and corrections are highlighted in green in the revised version of the manuscript.

I hope that revisions have improved the quality and the clarity of the paper and that it is now suitable for publication in Plos One.

Sincerely,

Dr Sonia Dhaouadi

National Observatory of New and Emerging Diseases, Tunis, Tunisia

sonidhaouadi88@gmail.com

Minor comments:

1. Page 4: In their rebuttal letter, the authors confirm my previous suspicion that the only stratification variable was region. However, now that the authors have further clarified their work, I think that the sampling procedure was also stratified by area of residence, as healthcare centers were selected from both urban and rural areas.

Authors’ response: Thank you for your comment, area of residence was also considered as suggested as variable of stratification (Methods, sampling procedure, line 122).

2. Page 4, Lines 126-127: The authors explain that patients were randomly selected at healthcare centers, but the authors may also want to explain how this was done in practice. For example, did the study staff enter the total number of patients on the patient list into a computer, which then returned a certain number of randomly selected patients, such as patients 2, 5, and 6 on the list?

Authors’ response: This comment was taken into account in Methods section, sampling procedure, lines 129-130

3. Page 5: The authors have not addressed my question of whether it is appropriate to ignore the sampling procedure in the analysis. If I am not mistaken, Appendix S1 contains all the information required to incorporate this information. (The percentages at each stage of the sampling procedure could be used to construct sampling weights, could they not?)

Authors’ response: Thank you for your comment, this proposition was taken into consideration in the results section, all percentages, crude ORs and their 95% confidence interval were weighted as mentioned according to region, governorate and area of residence.

Thank you again for taking the time to share your insightful feedback.

---

## [Editor Report · Decision Letter 3]

2 Mar 2022

Knowledge Attitudes and Practices toward seasonal influenza vaccine among pregnant women during the 2018/2019 influenza season in Tunisia

PONE-D-21-08415R3

Dear Dr. Bettaieb,

We’re pleased to inform you that your manuscript has been judged scientifically suitable for publication and will be formally accepted for publication once it meets all outstanding technical requirements.

Kind regards,

Ammal Mokhtar Metwally, Ph.D (MD)

Academic Editor

PLOS ONE
---

## [Editor Report · Acceptance letter]

14 Mar 2022

PONE-D-21-08415R3 

Knowledge Attitudes and Practices toward seasonal influenza vaccine among pregnant women during the 2018/2019 influenza season in Tunisia 

Dear Dr. Bettaieb:

I'm pleased to inform you that your manuscript has been deemed suitable for publication in PLOS ONE. Congratulations! Your manuscript is now with our production department. 

Kind regards, 

on behalf of

Professor Ammal Mokhtar Metwally 

Academic Editor

PLOS ONE